# Translating best practice into real practice: Methods, results and lessons from a project to translate an English sexual health survey into four Asian languages

Daniel Vujcich[1]*, Meagan Roberts[1], Zhihong Gu[2‡], Shih-Chi Kao[3‡], Roanna Lobo[1], Limin Mao[4‡], Enaam Oudih[5‡], Nang Nge Nge Phoo[1‡], Horas Wong[4,6‡], Alison Reid[1]

1 School of Population Health, Curtin University, Perth, Western Australia, Australia, 2 Ethnic Communities Council of Queensland, Brisbane, Queensland, Australia, 3 HIV and Related Programs Unit, Population Health, Sydney Local Health District, Sydney, New South Wales, Australia, 4 Centre for Centre for Social Research in Health, UNSW Sydney, New South Wales, Australia, 5 Relationships Australia, Adelaide, South Australia, Australia, 6 Kirby Institute, UNSW Sydney, New South Wales, Australia

☯ These authors contributed equally to this work.
‡ These authors also contributed equally to this work
* daniel.vujcich@curtin.edu.au

**Data Availability Statement:** All relevant data are within the paper and its Supporting information files.

## Abstract

### Background

Migrants are underrepresented in population health surveys. Offering translated survey instruments has been shown to increase migrant representation. While 'team translation' represents current best practice, there are relatively few published examples describing how it has been implemented. The purpose of this paper is to document the process, results and lessons from a project to translate an English-language sexual health and blood-borne virus survey into Khmer, Karen, Vietnamese and Traditional Chinese.

### Methods

The approach to translation was based on the TRAPD (Translation, Review, Adjudication, Pretesting, and Documentation) model. The English-language survey was sent to two accredited, independent translators. At least one bilingual person was chosen to review and compare the translations and preferred translations were selected through consensus. Agreed translations were pretested with small samples of individuals fluent in the survey language and further revisions made.

### Results

Of the 51 survey questions, only nine resulted in identical independent translations in at least one language. Material differences between the translations related to: (1) the translation of technical terms and medical terminology (e.g. HIV); (2) variations in dialect; and (3) differences in cultural understandings of survey concepts (e.g. committed relationships).

**Funding:** This project was funded by the Australian Research Council (https://www.arc.gov.au/), Curtin University (https://www.curtin.edu.au/), ShineSA (https://shinesa.org.au/), the Queensland Department of Health (https://www.health.qld.gov.au/), the Western Australian Department of Health (https://ww2.health.wa.gov.au/), the South Australian Department of Health (https://www.sahealth.sa.gov.au) and the Victorian Department of Health (https://health.vic.gov.au/). The funder Curtin University provided support in the form of salaries for authors DV, MR, AR, and RL and scholarship support to NP, but did not have any additional role in the study design, data collection and analysis, decision to publish, or preparation of the manuscript. The specific roles of these authors are articulated in the 'author contributions' section. The funders had no role in study design, data collection and analysis, decision to publish, or preparation of the manuscript.

**Competing interests:** The authors have declared that no competing interests exist.

## Conclusion

Survey translation is time-consuming and costly and, as a result, deviations from TRAPD 'best practice' occurred. It is not possible to determine whether closer adherence to TRAPD 'best practice' would have improved the quality of the resulting translations. However, our study does demonstrate that even adaptations of the TRAPD method can identify issues that may not have been apparent had non-team-based or single-round translation approaches been adopted. Given the dearth of clear empirical evidence about the most accurate and feasible method of undertaking translations, we encourage future researchers to follow our example of making translation data publicly available to enhance transparency and enable critical appraisal.

## Introduction

In 2019, international migrants living in Oceania, North America and Europe accounted for 21 percent, 16 percent and 11 percent of the total population in those regions respectively [1]. Yet, migrants are frequently under-represented in population-level health studies [2–5]. Moradi and colleagues regard the "[s]ystematic under-representation of migrants in epidemiological studies and surveys [as] a serious methodological issue introducing bias and causing lack of generalizability of the results" [6]. Consequently, our ability to use the available evidence to accurately identify priority areas and effectively design policies and programs is compromised [5, 7–9].

Language is one barrier to migrant participation in research, and offering translated survey instruments has been shown to increase migrant representation [6, 9]. However, survey translation is not a straightforward process. As Curtarelli and van Houten note, a good translation must "on the one hand, take into consideration the different social realities, cultural norms, and respondent needs . . . and, on the other hand, respect the questionnaire design and retain measurement properties" [10].

There are several approaches to translation. Forward-only translation (also known as 'direct' or 'one-for-one' translation) involves a single individual translating an instrument from one language (the source language) into a second language (the target language) [11]. While forward-only translation has the advantage of saving time and costs, it is considered problematic because it "involves a total dependence on the [single] translator's skill and knowledge, and often results in low validity and reliability" [12]. Forward-backward translation (also known simply as 'back translation') represents an attempt to overcome the risks inherent in relying on a single individual. In forward-backward translation, a second individual translates the target language instrument back into the source language; the original source language instrument and the back-translated source language instrument are then compared, and any discrepancies serve as indications of the need for further refinements of the translation [13]. However, a criticism of forward-backward translation is that it has the potential to focus too narrowly on the task of literal translation at the expense of ensuring that the translation captures the intended meaning of the survey item in a way that is clear and suitable for the intended audience [14]. For example, Behr cites an example in which 'care services' was forward translated into German as *pflegedienste* and back translated as 'care services' suggesting no error, when in fact the translated term "did not fit the questionnaire context since it is only used in the context of the ill and/or the elderly and is thus not fitting to general child care

services" [15]. Ozolins and colleagues have reported how some forward translators choose literal translations (despite their misgivings as to whether it actually captures the intended meaning) because they do not want their translation to be flagged as an 'error' by the back translator [16].

Consequently, there has been a growing call for more nuanced and layered approaches to instrument translation in which accredited translators, other people who speak both the source and target languages, survey researchers, and subject matter experts work together to produce translated surveys which: (1) capture the intended meaning of the source instrument; (2) reflect the cultural and contextual specificities of the target population; and (3) will facilitate meaningful comparisons of data across populations [13, 17–21]. Indeed, the *Guidelines for Best Practice in Cross-Cultural Surveys* recommend "a team translation approach for survey instrument production" noting that "[o]ther approaches, such as back translation, although recommended in the past, do not comply with the latest translation research" [22]. 'Team translation' (also known as 'committee translation') is considered preferable to other approaches on the basis that it enables people with complementary knowledge and expertise to work together to arrive at the best translation to ensure that survey items convey what they were intended to convey to the target audience [21, 22].

While team translation can assume a variety of forms, the approach known as TRAPD (Translation, Review, Adjudication, Pretesting, and Documentation) is the version endorsed in the *Guidelines for Best Practice in Cross-Cultural Surveys* [22]. Under the TRAPD model, two independent translations are produced and are then compared (item-by-item) by bilingual reviewers who possess study design and subject-matter knowledge, and work with an adjudicator to identify the 'best' translation for pretesting; each step is documented for transparency [21, 22].

In this article, we apply the TRAPD model to translate an English-language sexual health and blood-borne virus survey into four languages for migrants living in Australia. The aim of the study is to:

1. document how TRAPD can be applied in practice, including any challenges in its application;

2. provide examples of issues identified through the processes of team-based 'review and adjudication' and pretesting, which are key features of the TRAPD model;

3. offer guidance to future researchers who seek to use the TRAPD method; and

4. provide recommendations for further research priorities on the subject of survey translation.

The study makes an important contribution to the literature since there are relatively few published examples describing how TRAPD has been implemented in the context of survey research, despite it being the model endorsed in the *Guidelines for Best Practice in Cross-Cultural Surveys* [22]. Much of the available TRAPD literature relates to translations carried out as part of large and relatively well-resourced surveys, such as the European Social Survey [23–28]. However, as Sha and Lai have argued "[i]t is important to identify a viable translation process that can be adapted and tailored to the varying level of expertise and resources available" [29].

## Methods

Ethics approval for this study was obtained from the Curtin University Human Research Ethics Committee HRE2019-0395 and participants provided written consent. Data were analysed anonymously. An English-language self-administered paper survey was developed and

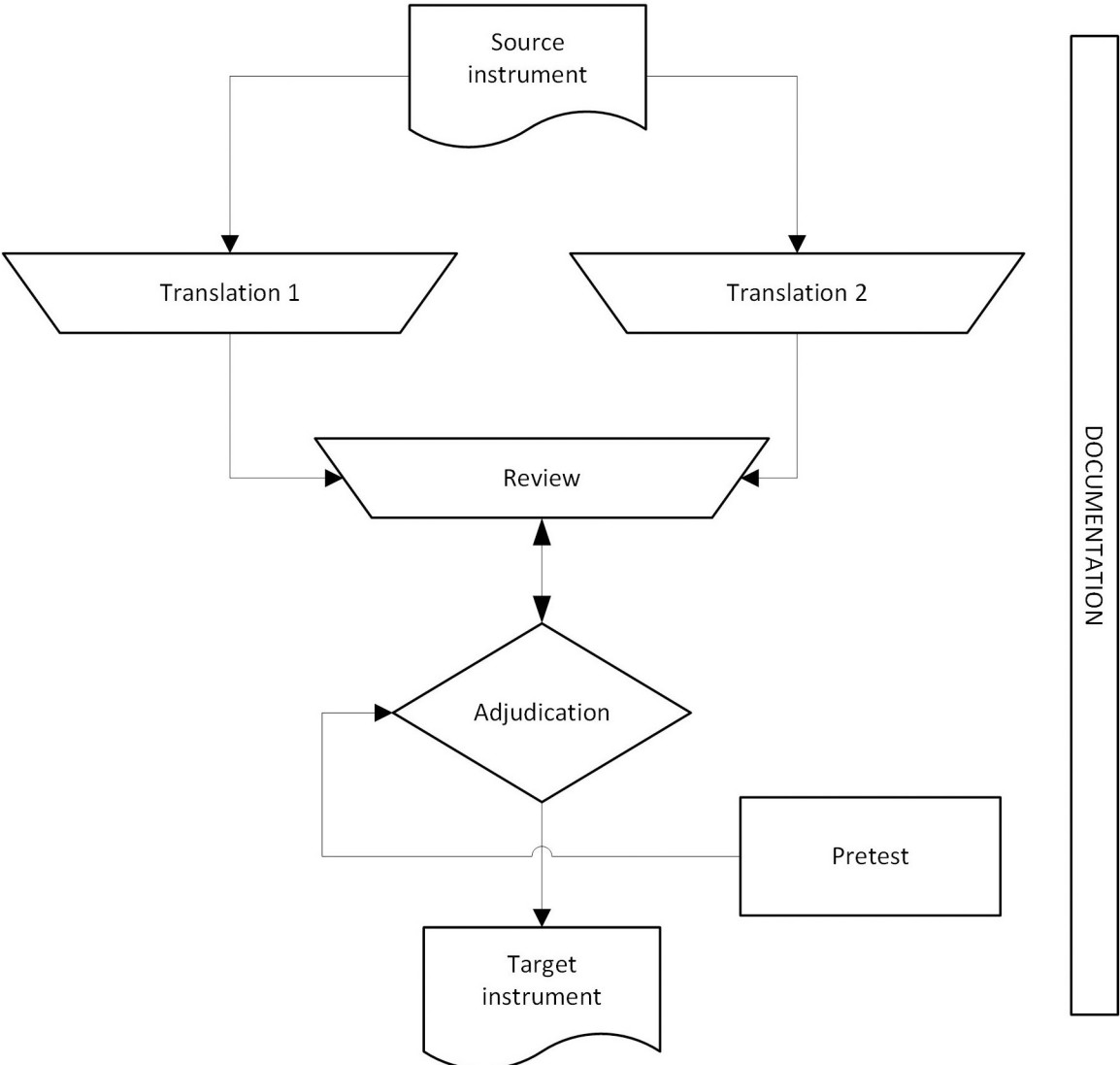

**Fig 1. Illustration of the TRAPD model adapted from the *Guidelines for Best Practice in Cross-Cultural Surveys* [22].**

pretested with South-East Asian, North-East Asian and Sub-Saharan African migrants living in Australia, as described elsewhere [30]. Due to budgetary constraints, the English-language survey could only be translated into a limited number of languages. As this was a feasibility study, four languages were chosen to gain insight into the complexities of multi-language survey development and administration. The selection of languages was informed by: (1) unpublished epidemiological data showing the incidence of sexually transmissible infections (STIs) and blood-borne viruses (BBVs), by country of birth; (2) estimated resident population data, by country of birth; (3) English-language proficiency data, by country of birth; and (4) input from community-based organisations working with migrant communities to improve STI and BBV outcomes. The resulting languages for translation (hereafter referred to as target languages) were Traditional Chinese, Karen, Khmer and Vietnamese.

The approach to translation was based on the TRAPD team translation model depicted in Fig 1, and described in the *Guidelines for Best Practice in Cross-Cultural Surveys* [22].

**Table 1. Example of information provided to translators.**

| Item | Annotation |
|---|---|
| Is there non-traditional medication available for people living with HIV so they can live a normal life? *(Tick one)* | The phrase "normal life" in this question is intended to indicate that the person can function like other people and do everyday things without being impeded by their HIV status. |
| | The phrase "non-traditional medication" is intended to refer to pharmaceuticals, as opposed to herbal or alternative medicines. |
| Can you get hepatitis B from swallowing food or water containing the faeces (poo) of an infected person? *(Tick one)* | There is no need to translate the informal 'poo' if people are likely to understand the translated term for 'faeces'. |
| If a close friend in Australia told you that they were going to get tested for STIs and BBVs, how would you feel? *(Tick any that apply)* | Here we are referring to a platonic (non-sexual) friendship |
| Since January 2018, have you had sex with a sex worker while overseas? *(Tick one)* | Please chose a neutral, non-stigmatising translation for the term 'sex worker'. |
| How do you identify? (Tick all that apply) | N/A |
| • Woman | Here we mean gender as opposed to biological sex |
| • Man | Here we mean gender as opposed to biological sex |
| • Non-binary person | Here we mean people who don't identify as either a man or woman |
| • Other (please specify): | N/A |
| • Transgender | N/A |
| • Cisgender | Here we mean people who identify with the gender they were assigned at birth |
| • Prefer not to answer | N/A |
| Which cultural / ethnic group categories do you identify with? | If Zulu and Hmong are not likely to be meaningful in the translated language please delete |
| For instance, you might identify with: | |
| • One group of people within your country of birth (e.g. Zulu, Hmong) | |
| • Your ancestral heritage (e.g. Indian-Malaysian or Chinese-Vietnamese) | |
| • People from another place you have lived (e.g. British) | |

## Translation

The English-language survey was sent to two accredited translation companies for independent translation. The companies were recommended by project partner organisations with experience working with the target populations. Each company was provided with a detailed brief which included contextual information about the survey, and general principles for translation based on *Guidelines for Best Practice in Cross-Cultural Surveys* (see S1 Appendix for example) [22]. Translators were asked to enter item-by-item translations into an annotated table comprising over 300 rows of questions, individual response options and instructions. The annotations provided extra information to assist translators to convey the intended meaning of the survey items. Some example annotations are reproduced in Table 1.

## Review

After the independent translations were completed, at least one bilingual person was chosen to review and compare the translations; where differences or errors were observed in the

**Table 2. Summary of reviewer characteristics, by language.**

| Language | Reviewer number | Place of birth | Years of residence in Australia | Field of occupation |
|---|---|---|---|---|
| Traditional Chinese | 1 | Mainland China | 24 | Community sector (health) |
| | 2 | Mainland China | 23 | Academia (health) |
| | 3 | Taiwan | 26 | Health planning |
| | 4 | Hong Kong | 6 | Academia (health) and clinical sexual health |
| Khmer | 1 | Cambodia | 10 | Community sector (health) |
| Vietnamese | 1 | Vietnam | 14 | Community sector (health) |
| Karen | 1 | Myanmar | 29 | Community sector (social work) |

translations, the role of the reviewer was to make recommendations about which translation best reflected the intended meaning of the relevant survey item.

An Excel spreadsheet was developed to assist with the review process (see S2 Appendix for Traditional Chinese example). The spreadsheet contained columns for the English survey item and both target language translations to facilitate side-by-side comparison of each item. Reviewers were instructed to use the spreadsheet to: (1) indicate whether any errors were present in either translation using a drop down menu based on the European Social Survey (ESS) Verifier Intervention Categories (S3 Appendix) [31]; (2) make any relevant additional comments about either translation; and (3) indicate a preferred translation for each item, with reasons.

Reviewers were generally from academic and community-sector organisations responsible for overseeing the design and implementation of the broader study and, as such, had knowledge of the survey context. The exception to this was the Karen reviewer who was recruited externally but had had previous experience with other sexual health research. A summary of the reviewers' background and experience is contained in Table 2.

The task of reviewing the translations was provided in-kind and, consequently, only one reviewer was recruited for Khmer, Vietnamese and Karen translations. In the case of Traditional Chinese, two reviewers from mainland China (third- and sixth-named authors) independently reviewed the translations and entered data into the spreadsheet. The reviewers recommended that individuals born in Hong Kong and Taiwan (fourth- and ninth-named authors) should also be involved in the review process as it was known that there were some linguistic variations for important terms.

In practice, only the Traditional Chinese reviewers used the ESS Categories. The Karen reviewer simply indicated a preferred translation for *each* survey item, usually with reasons. In the case of the Khmer and Vietnamese translations, the reviewers indicated that they preferred one translated version (in its entirety) over another. The implications of these variations are considered in the Discussion section below.

## Adjudication

As there was only one reviewer for the Khmer, Vietnamese and Karen translations, adjudication took the form of the first-named author assessing the reviewers' comments and accepting recommendations unless they did not appropriately convey the intended meaning of the source text (examples of where this occurred are set out below). For the Khmer and Vietnamese translations, no adjudication was possible as reviewers' did not provide reasons for preferring one translation over another; instead, the reviewers' recommendations were accepted and the preferred translation was sent for pretesting.

## Pretesting

Ethics approval for pretesting was obtained (Curtin University Human Research Ethics Committee 2019–0395). The minimum target sample size for the pilot survey (in any language) was 1,116 respondents equally divided between the three regions of birth (Sub-Saharan Africa, North East Asia and South-East Asia) to detect regional differences at a significance level of 5% and 90% power.

Given the limited availability of resources for pretesting, the exploratory nature of the feasibility study, and the fact that multiple quality checking methods were built into the study design (e.g. two independent translations, review and adjudication), a pragmatic decision was made to only pretest translations on small samples. Pretest participants were recruited using convenience sampling by two members of the research team who had experience working with these migrant communities. The size of the samples varied by community—Chinese (n = 3), Vietnamese (n = 20), Khmer (n = 4) and Karen (n = 3). The larger sample size for Vietnamese pretesting was opportunistic in the sense that a group of 20 participants were gathered for another purpose and expressed willingness to provide feedback on the translated instrument; although this resulted in the Vietnamese pretest sample being larger than those representing other language groups, the opportunity to obtain more feedback with minimal additional resources was recognised as an efficient means of obtaining more data to check instrument validity. There was no intention to engage in any statistical comparison of differences in pretest responses between the communities. While all pretest participants were fluent in the survey language being pretested, no other demographic characteristics were recorded.

Pretesting was conducted in groups in which participants were asked to complete the draft translated surveys and then answer the following questions:

- Was there anything that you did not understand? If so, what was it and why do you think you had trouble?

- Did you find it difficult to answer any questions? If so, which questions and why?

- Were there any errors in the survey that you noticed?

Written notes summarising the participants' responses were prepared by the pretest facilitators and sent to the adjudicator.

## Results

S3 Appendix compares the independent translations for each survey item in each language. As summarised in Table 3, of the 51 survey questions (including five Likert statements), only nine resulted in identical independent translations in at least one language.

While identical independent translations were rare, many of the differences in translation were not material in the sense that they did not change the intended meaning of the source text. For instance, the English source item–*Which of the following best describes you*? *(Tick one)*–was translated into Traditional Chinese as follows:

Translation 1: 以下哪一項陳述最符合你的情況?(勾選一項) Which one of the following best describes you? (Tick one) (*informal 'you')

Translation 2: 以下哪項最符合您的情況?(勾選一項) Which of the following best describes you? (Tick one) (*formal 'You')

Similarly, the reviewers considered both Traditional Chinese translations of "How old are you?" to be interchangeable, the main difference being tone (colloquial versus formal):

**Table 3. Identical independent translations of substantive English-language survey questions, by language of translation.**

| Source text (English) | Whether independent translations were identical (Yes/No) | | | |
|---|---|---|---|---|
| | Khmer | Traditional Chinese | Karen | Vietnamese |
| *Is an HIV test done whenever someone has a blood test in Australia?* | No | No | No | No |
| *Can hepatitis C be passed on by sharing injecting equipment like needles and syringes?* | Yes | No | No | No |
| *Did you use a condom the MOST RECENT time you had sex?* | No | No | No | No |
| *Why did you NOT use a condom the most recent time you had sex?* | No | No | No | No |
| *If a close friend in Australia told you that they were going to get tested for STIs and BBVs, how would you feel?* | No | No | No | Yes |
| *How old are you?* | No | No | Yes | Yes |
| *What is your religion?* | No | No | No | Yes |
| *What are the main languages you speak at home?* | No | No | No | No |
| *[Statements for Likert-scale responses]* | | | | |
| • *I felt upset\** | No | No | No | Yes |
| • *I felt embarrassed* | Yes | No | No | No |
| • *The survey was too long* | Yes | No | No | No |
| • *I found it hard to understand some questions / words* | Yes | No | No | No |

Translation 1: 你幾歲 (How old are you)?

Translation 2: 您的年齡是 (What is your age)?

However, a number of more material differences between translations were detected during the review process, a sample of which is highlighted in Table 4 below. In most instances, one translator's version was preferred over another; however, there were instances in which the reviewers and adjudicator considered that neither version suitably captured the intended meaning of the item.

Some differences in translation were the products of clear misunderstandings of the original English survey items. For instance, in the case of Khmer, Translator 1 interpreted 'committed relationship' to mean the equivalent of 'someone you have a breakup with', and 'casual sexual partner' was translated to mean 'unprotected sexual partner'. Neither of these translations reflected the intended meaning of the original English survey item and, in both cases, Translator 2's versions were preferred during the review process.

Other differences related to variations in dialect. A reviewer noted that one Karen translator used the dialect associated with the people of Karen State while the other used the dialect of the Irrawaddy delta region. These differences related to only a few survey response items and the adjudicator chose to adopt the dialect associated with the people of Karen State based on what was known about the profile of Karen migrants in Australia. In the context of Traditional Chinese, both translations adopted the dialect commonly spoken in mainland China and Hong Kong but the reviewers recommended that, in some cases, other dialects should be incorporated into the survey. For instance the reviewers noted that while the term 衣原體 (chlamydia) was familiar to people living in mainland China and Hong Kong, a different term was used in Taiwan—namely, 披衣菌. Similarly, reviewers noted that hepatitis B was written as 乙型肝炎 in mainland China and Hong Kong but B型肝炎 in Taiwan. Given that participants from both regions were expected to complete the Traditional Chinese version of the survey, the review panel determined that both translations should be included for completeness, e.g. in the case of chlamydia 衣原體(披衣菌).

A third category of differences related to technical terms. Vietnamese Translator 1 translated chlamydia as *bệnh hoa liễu* (a generic term for venereal diseases), and Translator 2

**Table 4. Examples of material differences between translations detected during the review process, and outcome of adjudication.**

| Items / key terms | Language | Translation 1 | Translation 2 | Preferred translation |
|---|---|---|---|---|
| Chlamydia | Vietnamese | Bệnh hoa liễu | Bệnh lậu (khuẩn Chlamydia) | Bệnh Chlamydia |
| | | *English meaning: Venereal disease* | *English meaning: Gonorrhoea (chlamydia bacteria)* | |
| | Traditional Chinese | 衣原體 | 衣原體病 | 衣原體 (披衣菌) 感染 |
| | | *English meaning: Chlamydia* | *English meaning: Chlamydia infection/sickness* | Chlamydia in two dialects + term for infection |
| Gonorrhoea | Vietnamese | Bệnh lậu | Bệnh da liễu | Translation 1 |
| | | *English meaning: Gonorrhoea* | *English meaning: Venereal disease* | |
| HIV | Traditional Chinese | 愛滋病病毒 | HIV | Combine—愛滋病病毒 (HIV) |
| | | *English meaning: AIDS virus* | *English meaning: HIV* | |
| | Karen | တၢ်ဆါဃၢ်ဒုးကဲထိၣ်အ�ွး (စ)တၢ်ဆါ (HIV) | HIV | Translation 1 |
| | | *English meaning: Human Immunodeficiency Virus (HIV)* | *English meaning: HIV* | |
| Questions about sexual activities | Karen | တၢ်သံကွၢ်ဘၣ်ဃးသ့ၣ် ထံးအတၢ်ဟူးတၢ်ဂဲၤတ ဖၣ်ဒီးတၢ်ရူလိာ်မုာ်လိာ် | တၢ်သံကွၢ်တဖၣ်ဘၣ်ဃးမုာ်ခွါ သ့ၣ်ထံးအတၢ်ဟူးတၢ်ဂဲၤဒီး တၢ်ရူလိာ်သးသ့ၣ်တဖၣ် | Translation 1 |
| | | *English meaning: Questions about sex* | *English meaning: Questions about sex among men and women* | |
| A casual sex partner | Vietnamese | Một người bạn tình không thường xuyên | Bạn tình bình thường (không có mối quan hệ ràng buộc) | Translation 2 |
| | | *English meaning: An occasional lover / a lover, not regular* | *English meaning: Casual partner / normal lover (no relationship tie)* | |
| | Traditional Chinese | 臨時性伴侶 | 隨意的性伴侶 | 非固定性伴(沒有穩定關係或臨時性伴, 包括一夜情人) |
| | | *English meaning: Temporary partner* | *English meaning: sex partner* | *Non-regular sexual partners (no stable relationships or temporary sexual partners, including one-night lovers)* |
| | Khmer | ដៃគូរួមភេទដែលមិនមានទំនាក់ទំនងច្បាស់លាស់ជាប្រចាំ | ដៃគូរួមភេទមួយគ្រាប់ | Translation 2 |
| | | *English meaning: Unprotected sexual partner* | *English meaning: Occasional sexual partners* | |
| Someone you are in a committed relationship with (e.g. husband / wife, boyfriend / girlfriend) | Karen | ပုၤတဂၤလၢနအၢၣ်လီၤ လၢနကအိၣ်ယှာ်အီၤလၢ သးတၢ်နီၢ် (အဒိ - ဝၤ,မါ,တၢ်အဲၣ်တီခွါ,တၢ် အဲၣ်တီမုၣ်) | ပုၤတဂၤဂၤလၢနအိၣ်ဒီးတၢ်ဘ ၣ်ထွဲလၢအအိၣ်ဒီးတၢ်အၢၣ်လီ ၤအီၤ (အဒိ - မါ / ဝၤ, တၢ်အဲၣ်တီမုၣ် / တၢ်အဲၣ်တီခွါ) | Translation 2 |
| | | *English meaning: 'Committed relationship' translated as 'willing—in mind- to live together'* | *English meaning: 'Committed relationship'* | |
| | Khmer | នរណាម្នាក់ដែលអ្នកមានទំនាក់ទំនងបូជ្រកំផ្ដាច់ជាមួយ (ឧទាហរណ៍ ប្ដី / ប្រពន្ធ មិត្តប្រុស / មិត្តស្រី) | មនុស្សម្នាក់ដែលអ្នកមានទំនាក់ទំនងជាមួយច្បាស់លាស់ (ឧទាហរណ៍ ប្ដី / ប្រពន្ធ, មិត្តប្រុស/មិត្តស្រី) | Translation 2 |
| | | *English meaning: Someone you have a breakup with (for example, spouse / boyfriend / girlfriend)* | *English meaning: Someone you have a clear relationship with (e.g. spouse, boyfriend / girlfriend)* | |

(*Continued*)

**Table 4.** (Continued)

| Items / key terms | Language | Translation 1 | Translation 2 | Preferred translation |
|---|---|---|---|---|
| Did you use a condom the MOST RECENT time you had sex? (Tick one) | Karen | မှၢ်နသူ၀ဲဒၣ်ဒၣ်တၢ်ဖှိၣ်(condom)တခါဖဲနအိၣ်ဃုာ်လိာ်သးလၢခံကတၢၢ်တဘှီအခါန့ၣ်ဧါ. | ဖဲနမံဃုာ်အလီၢ်ခံကတၢၢ်တဘှီနၣ်,မှၢ်နစူးကါထ့ၣ်ဖှိၣ်စ့ၢ်ကီးဧါ. (တိၤနီၣ်ဃာ်တခါ) | Translation 2 |
| | | English meaning: Condom translated as cover of penis | English meaning: Condom translated as cover of penis | |
| How old are you? | Karen | အသးအိၣ်ထဲလဲၣ်. (တိၤနီၣ်တခါ) | နသးနံၣ်အိၣ်ဝဲနံၣ်လဲၣ်. (တိၤနီၣ်ဃာ်တခါ) | Translation 2 |
| | | Dialect of Irrawaddy Region | Dialect of Karen State | |

translated it as *bệnh lậu (khuẩn Chlamydia)* which literally means "*gonorrhoea (Chlamydia bacteria)*". Neither translation was considered suitable, leading the reviewer to suggest a third term *(Bệnh Chlamydia)* which was adopted. However, there were instances in which reviewers determined that a balance needed to be struck between technical accuracy and comprehensibility for lay audiences. An example of this related to the translation of Human Immunodeficiency Virus (HIV) into Traditional Chinese. In one translation HIV was translated to 愛滋病病毒 (AIDS virus), while in the other it was translated using the more clinically accurate 人類免疫缺陷病毒 (HIV). A reviewer noted that:

> HIV is not AIDS virus. However, many people don't know what 人類免疫缺陷病毒(HIV's correct [or technical] name in Chinese) is but all know 愛滋病病毒 (translation of AIDS virus). In addition, not all people know HIV if we use the term in English instead of the translation. This translation 愛滋病病毒 is well known and used by the media and the community in their hometown. For the survey's purpose I think it is better to use 愛滋病病毒.

Through the review and adjudication process, a compromise was reached whereby the more widely understood translation (AIDS virus) was used with the more technically accurate English abbreviation (HIV) in parentheses.

Finally, some differences in translations reflected different cultural understandings. In relation to the English section heading "Questions about sexual activities and relationships", one Karen translator interpreted "sexual activities" as "sex among men and women". Notwithstanding the reviewer's preference for this translation, the adjudicator adopted the alternative translation (which was not limited to heterosexual activities) given that the survey was intended to capture the full spectrum of sexual relationships. Additionally, the Karen reviewer preferred to translate 'condom' as 'the cover of this' rather than 'the cover of the penis' on the basis that it was less direct and offensive. However, after consultation with the reviewer, the adjudicator opted for the latter on the basis that it was less euphemistic and thus more easily understood. Both the adjudicator and the reviewer agreed not to use the Karen translation which equated the concept of "committed relationship" with a term meaning "willing—in mind—to live together" on the basis that it did not capture the meaning of the source item which was intended to include partners who were not co-habiting.

Through pretesting, two further revisions to the Vietnamese translation were proposed and adopted. The first related to the translation for "non-traditional medicine" (*thuốc phi truyền thống*). Participants expressed the view that *thuốc Tây* (Western medicine) would be easier to understand. The second revision related to the translation for the question "What is the post-code in which you live?" Participants felt that the Vietnamese term for postcode (*mã bưu điện)* was unfamiliar and recommended that the English term should be added in parentheses to improve understanding: *Mã bưu điện (postcode) nơi bơn sống là gì*?

No significant issues were identified when pretesting the Traditional Chinese translation. Similarly, pretesting of the Khmer survey did not yield any substantive changes; the changes were limited to some grammatical amendments to the introductory text and the detection of minor formatting errors.

Two of the Karen pretesters said that they did not have any comments to make. A third Karen-speaker with experience with translations detected some minor errors around format-ting (e.g. some lower portions of the script was not visible for some words) and identified some typographical errors. The main substantive change related to the translation of HIV: "It is translated as human immunodeficiency virus, so many people would find it hard to under-stand. It will be better if the pronunciation of HIV is written in Karen (as it was done for the country names, gonorrhoea, syphilis, chlamydia)". Additionally, this pretesting participant noted that the character signifying 'no/not' had been omitted from one translation which fun-damentally changed the intended meaning. All suggested changes were adopted.

## Discussion

Whereas much of the existing TRAPD literature reports on the use of the method in the con-text of large and relatively well-resourced, international surveys, this study documents the experience of applying the TRAPD method to a modestly-resourced survey in one country. The purpose of this study is not to provide evidence that TRAPD is the final word in effective translation methods. As has been noted elsewhere, empirical data on "the contribution of the TRAPD process . . . to the overall quality of the translated survey instrument" [18] remains limited, and more studies comparing the results of different translation methods are needed. Instead, this paper shows that the TRAPD method has value in identifying errors and exposing nuances in translation, but that it can be difficult to implement in practice, particularly where resources are low and translations into many languages are required. As Knight and colleagues acknowledge, the reality of studying minority populations is that "research context often makes full compliance with . . . best practice recommendations impossible and . . . the best one can do is to make incremental approximations of these recommendations" [32].

As was reported in the Methods, deviations from TRAPD 'best practice' occurred. Signifi-cantly, it was not possible to employ or compensate translation reviewers; instead, the task of reviewing was undertaken in-kind which made it difficult both to recruit multiple reviewers for each language, and to rigidly proscribe the manner in which the translation reviews were undertaken. For instance, while reviewers were instructed to code translation errors using the European Social Survey (ESS) Verifier Intervention Categories, few reviewers used the ESS cat-egories, and some did not document any reasons for preferring one translation over another which made the process of adjudication difficult. Problems around documentation in the TRAPD review process have been noted in other studies, including the European Social Survey where it was noted that "maintaining documentation can be burdensome . . . The documenta-tion provided by the country teams on the development of ESS translations was at times mea-gre" [21].

In addition to being time-consuming, the practice of coding translation errors may not be familiar to bilingual reviewers who do not necessarily possess a research background. Many of the distinctions between the categories—for instance, register/wording issues, grammar/syntax issues and minor linguistic defects (which can include grammar)–are subtle or require a high level of linguistic awareness. This suggests that the use of alternative coding protocols should be considered and tested, and efforts should be made to simplify the process of documenting reasons for decision at each stage of the translation process. In an effort to understand the difficulties they experienced in obtaining adequate documentation from translation partners, Harkness, Villar and Edwards surmised "[t]his may be because those involved were not familiar with how and what to document, but it is also likely that the effort involved in manual documentation played a role" [21]. One solution may be for a member of the research team to assume responsibility for documentation by sitting with each panel member (either physically or virtually) while they independently review the translations. The researcher could record the preferred translation for each item and the reviewer's reasons, using probing questions where necessary; this was the approach taken for the Karen translation in our study. An alternative would be to investigate the development and use of specialised software to manage the documentation process. It is noted that an online, interactive Translation Management Tool was developed for the Survey of Health, Ageing and Retirement in Europe (SHARE) and tested in round 8 of the European Social Survey in 2016, although a number of areas for improvement were identified [21]. Currently, the tool is not publicly available but can be tailored to support specific survey requirements at a cost.

It is not possible to determine whether closer adherence to TRAPD 'best practice' in this study would have improved the quality of the resulting translations. A literature review of studies using multi-step, team-based translations of health quality of life questionnaires found that "[a]lthough there is some evidence that different methods (i.e., 'light/simpler' vs. 'heavy/complex') yield similar results, this has been tested empirically only on a very limited scale" [12]. However, our study does demonstrate that even 'light/simpler' adaptations of the TRAPD method can successfully identify issues that may not have been apparent had non-team-based or single-round translation approaches been adopted. For instance:

- relying on only one translator to conduct a forwards-only translation without further review would have resulted in important errors being overlooked (e.g. the Khmer translation of 'casual sexual partner' as 'unprotected sexual partner', and the Karen translation of 'committed relationship' as 'willing in mind to live together');

- relying on forward-backward translation only may not have revealed importance nuances (e.g. the translation of hepatitis B as 乙型肝炎 would likely have been translated back into English as 'hepatitis B' suggesting no problem with the translation, when in fact the review process highlighted that it would not be inclusive of/familiar to Taiwanese-Chinese speakers);

- not including researchers as partners in the translation process may have resulted in translations which did not reflect the intended meaning of the survey items (e.g. a bilingual reviewer without a sexual health and blood-borne virus background favoured the narrow, heteronormative translation of Karen translation of 'sexual activities' which meant 'sex among men and women');

- pretesting assisted in identifying typographical/formatting errors made when incorporating the reviewed translations into the survey tools (e.g. the omission for the term 'no/not' in one item in the Karen survey, and parts of the Karen script not being displayed correctly);

- the benefits of assembling a team comprising people with complementary expertise was particularly evident when attempting to translate technical, medical terminology in a way that, on the one hand, is meaningful to the target audience and, on the other hand, guards against perpetuating terms which are stigmatising or give rise to inaccurate assumptions (e.g. the difference between 'clinical' and 'lay' translations of HIV into Traditional Chinese).

It is, of course, possible that some errors in translation were not detected as a result of the approach we adopted. For this reason, we have taken the rare step of making all of our translation data publicly available (S3 Appendix). Given the dearth of clear empirical evidence about the most accurate and feasible method of undertaking translations, we encourage future researchers to follow our example. Greater transparency around survey translation research processes and results will: (1) allow the accuracy of different approaches to be studied in greater detail; (2) provide practical insight into effective ways of adapting established processes in response to different needs, settings and levels of resourcing; and (3) enable the validity of the survey results to be assessed with reference to the quality of the underlying translations.

Unless a culture of transparent trial-and-error in translation is fostered, negative consequences will follow. First, survey researchers may feel daunted by the complexity (and uncertainty) of translation and may therefore avoid attempts to engage with linguistically diverse populations; this would either result in: (a) underrepresented sections of the population continuing to be overlooked in studies geared towards collecting data to improve health outcomes, or (b) respondents having no option but to complete surveys in a language other than their preferred language, thus increasing the potential for survey items to be misunderstood and data quality to be compromised. Second, survey researchers who have attempted to undertake translations but whose methods have deviated from 'best practice' (e.g. due to circumstantial challenges like budget limitations) may hesitate to publish their translation processes and results for fear of being criticised; the consequence of this would be a continued dearth of information about the most effective and efficient means of undertaking survey translations. As a recent editorial in *Nature Human Behaviour* makes plain, "[s]cience is messy, and the results of research rarely conform fully to plan or expectation. 'Clean' narratives are an artefact of inappropriate pressures and the culture they have generated" [33]. It is only through more detailed accounts of the processes and results of survey translation that we can come closer to understanding which methods (and variations of methods) are best suited to produce the most accurate results in specific contexts. Given the increase in the scale of international (pre-pandemic) migration [34], and the growing appetite for multinational, multicultural and multiregional surveys [35], the importance of continuing to build the evidence-base for survey translation is both apparent and urgent.

## Conclusions

Survey translation is time-consuming and costly and, as a result, deviations from TRAPD 'best practice' occurred. It is not possible to determine whether closer adherence to TRAPD 'best practice' would have improved the quality of the resulting translations. However, our study does demonstrate that even adaptations of the TRAPD method can identify issues that may not have been apparent had non-team-based or single-round translation approaches been adopted. Given the dearth of clear empirical evidence about the most accurate and feasible method of undertaking translations, we encourage future researchers to follow our example of making translation data publicly available to enhance transparency and enable critical appraisal.

## Supporting information

**S1 Appendix. Brief for Translators.**
(DOCX)

**S2 Appendix. Reviewers' spreadsheet (Chinese).**
(XLSX)

**S3 Appendix. TRAPD results.**
(XLSX)

## Acknowledgments

We acknowledge the independent translators from Aussie Translations and Global Village, and the community participants who assisted with the review and pretesting processes. We are also grateful for the assistance of Ms Caitlin Wilshin in assisting to prepare this manuscript for submission.

## Author Contributions

**Conceptualization:** Daniel Vujcich, Roanna Lobo, Alison Reid.

**Data curation:** Daniel Vujcich, Meagan Roberts, Zhihong Gu, Shih-Chi Kao, Limin Mao, Enaam Oudih, Nang Nge Nge Phoo, Horas Wong.

**Formal analysis:** Daniel Vujcich, Zhihong Gu, Shih-Chi Kao, Roanna Lobo, Limin Mao, Enaam Oudih, Nang Nge Nge Phoo, Horas Wong, Alison Reid.

**Funding acquisition:** Roanna Lobo, Alison Reid.

**Investigation:** Daniel Vujcich, Zhihong Gu, Shih-Chi Kao, Roanna Lobo, Limin Mao, Nang Nge Nge Phoo, Horas Wong, Alison Reid.

**Methodology:** Daniel Vujcich, Roanna Lobo, Alison Reid.

**Project administration:** Daniel Vujcich, Meagan Roberts, Roanna Lobo, Alison Reid.

**Supervision:** Zhihong Gu, Roanna Lobo, Enaam Oudih, Alison Reid.

**Validation:** Daniel Vujcich, Meagan Roberts, Zhihong Gu, Shih-Chi Kao, Limin Mao, Horas Wong.

**Writing – original draft:** Daniel Vujcich.

**Writing – review & editing:** Daniel Vujcich, Meagan Roberts, Zhihong Gu, Shih-Chi Kao, Roanna Lobo, Limin Mao, Enaam Oudih, Nang Nge Nge Phoo, Horas Wong, Alison Reid.

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
