## [Editor Report · Decision Letter 0]

2 Jul 2021

PONE-D-21-17384

Translating best practice into real practice: methods, results and lessons from a project to translate an English sexual health survey into four Asian languages

PLOS ONE

Dear Dr. Vujcich,

Thank you for submitting your manuscript to PLOS ONE. After careful consideration, we feel that it has merit but does not fully meet PLOS ONE’s publication criteria as it currently stands. Therefore, we invite you to submit a revised version of the manuscript that addresses the points raised during the review process.

ACADEMIC EDITOR COMMENTS:

Thank you for inviting me to review this manuscript. The manuscript is well written. Actually, it's challenging to translate survey or questionnaire into different languages with different context and culture. I have a few questions and comments for the authors which are listed in the following:

1. The team approach TRAPD for translation is systematic, but no statistical method conducted to assess the equivalence such as content validity test (CVI). Is it possible to add or combine statistical method to increase the validity of the translation?

2. In the pretesting phase, the sample size of Vietnamese was 20, but other migrant communities were 3 or 4. Why was there a great variation in the sample size in different groups?

3. In line 141-142, "While identical independent translations were rare, many of the differences in translation were not material." What's the meaning of "material"?

4. In the Discussion, it is stated that "Brislin's approach to back-translation was regarded as the gold standard". What's the justification for adopting other approach e.g. TRAPD for translation but not adopting Brislin's approach? What's the advantages of TRAPD over Brislin's approach?

We look forward to receiving your revised manuscript.

Kind regards,

Ka Ming CHOW

Academic Editor

PLOS ONE

Journal Requirements:

2. In your Methods section, please provide a justification for the sample size used in your study during the pre-testing of the translated questionnaire, including any relevant power calculations (if applicable).

3. Please provide additional details regarding participant consent. In the ethics statement in the Methods and online submission information, please ensure that you have specified whether consent was written or verbal/oral. If consent was verbal/oral, please specify: 1) whether the ethics committee approved the verbal/oral consent procedure, 2) why written consent could not be obtained, and 3) how verbal/oral consent was recorded. If your study included minors, please state whether you obtained consent from parents or guardians in these cases. If the need for consent was waived by the ethics committee, please include this information.

"This project was funded by the Australian Research Council (https://www.arc.gov.au/), Curtin University (https://www.curtin.edu.au/), ShineSA (https://shinesa.org.au/), the Queensland Department of Health (https://www.health.qld.gov.au/), the Western Australian Department of Health (https://ww2.health.wa.gov.au/), the South Australian Department of Health (https://www.sahealth.sa.gov.au) and the Victorian Department of Health and Human Services (https://www.dhhs.vic.gov.au/). 

We note that one or more of the authors is affiliated with the funding organization, indicating the funder may have had some role in the design, data collection, analysis or preparation of your manuscript for publication; in other words, the funder played an indirect role through the participation of the co-authors. If the funding organization did not play a role in the study design, data collection and analysis, decision to publish, or preparation of the manuscript and only provided financial support in the form of authors' salaries and/or research materials, please do the following:

a. Review your statements relating to the author contributions, and ensure you have specifically and accurately indicated the role(s) that these authors had in your study. These amendments should be made in the online form.

b. Confirm in your cover letter that you agree with the following statement, and we will change the online submission form on your behalf: 

“The funder provided support in the form of salaries for authors [insert relevant initials], but did not have any additional role in the study design, data collection and analysis, decision to publish, or preparation of the manuscript. The specific roles of these authors are articulated in the ‘author contributions’ section.

6. Please remove your figures from within your manuscript file, leaving only the individual TIFF/EPS image files, uploaded separately.  These will be automatically included in the reviewers’ PDF.

---

## [Author Response · Author response to Decision Letter 0]

19 Jul 2021

Dear Editors, 

Translating best practice into real practice: methods, results and lessons from a project to translate an English sexual health survey into four Asian languages 

Thank you for your considered and helpful feedback on our manuscript. We have reproduced and responded to each item of feedback below. 

1. ASSESSING EQUIVALENCE 

Feedback:

• The team approach TRAPD for translation is systematic, but no statistical method conducted to assess the equivalence such as content validity test (CVI). Is it possible to add or combine statistical method to increase the validity of the translation?

Response: 

While statistical techniques such as exploratory and confirmatory factor analysis are commonly used to test equivalence of clinical assessment instruments and screening tools, they are less common in the context of population surveys geared towards measuring knowledge, attitudes and behaviours. The Cross-Cultural Survey Guidelines state that: “When multiple languages are used in the same survey, pretesting the different language versions is an essential part of ensuring measurement equivalence and cultural and cross-cultural equivalence” (emphasis added). Pretesting data are typically collected using qualitative methods such as cognitive interviews, focus group discussions and behaviour coding. 

Assessing equivalence through the use of qualitative methods is the approach taken in a number of population surveys, including: 

• The National Health Interview Survey (Cancer Control Module) and the California Health Interview Survey (Willis G, Stapleton M.S., Leven K., Norberg A., Stark D., Forsyth B., Brick P.D., Berrigan D., Thompson F.E., Lawrence D., Hartman A. Evaluation of a multistep survey translation process. In Harkness J., et al (eds). Survey methods in multinational, multiregional, and multicultural contexts. London, John Wiley & Sons. 2010); 

• The Tobacco Use Survey component of the Current Population Survey (Forsyth B., Kudela M.S., Levin K., Lawrence D., Willis G. Methods for translating an English-language survey questionnaire on tobacco use in Mandarin, Cantonese, Korean, and Vietnamese. Field Methods. 2007;19(3):264-283); 

• US Household Food Security Survey (Kwan C.M., Napoles A.M., Chou J., Seligman H.K. Development of a conceptually equivalent Chinese-language translation of the US Household Food Security Survey Module for Chinese immigrants to the USA. Public Health Nutr. 2015;18(2):242-250); 

• An 11-language survey conducted by the European School on Interdisciplinary Tinnitus Research (Biswas R., Lugo A., Gallus S., Akeroyd M.A., Hall D.A. Standardized questions in English for estimating tinnitus prevalence and severity, hearing difficulty and usage of healthcare resources, and their translation into 11 European languages. Hear. Res. 2019;377:330-338); and 

• The Parent Attitudes about Childhood Vaccines Survey (Cunningham R.M., Kerr G.B., Orobio J., et al Development of a Spanish version of the parent attitudes about childhood vaccines survey. Hum. Vaccines Immunother. 2019;15(5):1106-1110). 

In our Discussion we note that: 

… [A]dditional and innovative techniques for quality appraisal are prudent to check the quality of translations. For instance, in the context of our study, we plan to compare results obtained from English-language and target-language surveys, controlling for factors such as country of birth, length of time in Australia and gender. Any differences in responses may serve as an indication that the translated version did not convey the same meaning as the English version of the survey, prompting further investigation into the adequacy of the translation.

However, we consider that actually undertaking this analysis lies beyond the scope of this manuscript, the explicit aim of which is to document the process and results of applying the conventional TRAPD method. 

2. SAMPLE SIZE

Feedback: 

• In the pretesting phase, the sample size of Vietnamese was 20, but other migrant communities were 3 or 4. Why was there a great variation in the sample size in different groups?

• In your Methods section, please provide a justification for the sample size used in your study during the pre-testing of the translated questionnaire, including any relevant power calculations (if applicable).

Response: 

Decisions around sample size for pretesting must be read in context. In the Introduction, we state: “The project formed one part of study to ascertain the feasibility of conducting a periodic national survey in Australia to measure migrants’ knowledge, attitudes and practices in relation to sexual health and blood-borne viruses.” We have further clarified the study’s emphasis on investigating feasibility in the first paragraph of the Methods section: 

“Due to budgetary constraints, the English-language survey could only be translated into a limited number of languages. As this was a feasibility study, four languages were chosen to gain insight into the complexities of multi-language survey development and administration”. 

Additionally, we have redrafted the pretesting sub-section of the Methods to make it clear that the sampling strategy for pretesting was pragmatic and primarily guided by the availability of resources: 

The minimum target sample size for the pilot survey (in any language) was 1,116 respondents equally divided between the three regions of birth (Sub-Saharan Africa, North East Asia and South-East Asia) to detect regional differences at a significance level of 5% and 90% power. Based on available Census data on country of birth and English-language proficiency, it was estimated that:

• 82.00% (n=305) of North East Asian-born respondents would be from a country in which Chinese was the dominant language and that, assuming all spoke some Chinese, 25.9% (n=79) would be likely to not speak English well or at all; 

• 25.21% (n=94) of South East Asian-born respondents would be from Vietnam and that, assuming all spoke Vietnamese, 31.50% (n=30) would be likely to not speak English well or at all;

• 3.81% (n=14) of South East Asian-born respondents would be from Cambodia and that, assuming all spoke Khmer, 33.50% (n=5) would be likely to not speak English well or at all;

• 3.75% (n=14) of South-East Asian-born respondents would be from Myanmar and less than 10% would speak Karen (proportion of those who would be likely to speak English not well or not at all is unknown) [20,21]. 

Given these small numbers, the limited availability of resources for pretesting, the exploratory nature of the feasibility study, and the fact that multiple quality checking methods were built into the study design (e.g. two independent translations, review and adjudication), a pragmatic decision was made to only pretest translations on small samples. Pretest participants were recruited using convenience sampling by two members of the research team who had experience working with these migrant communities. The size of the samples varied by community – Chinese (n=3), Vietnamese (n=20), Khmer (n=4) and Karen (n=3). The larger sample size for Vietnamese pretesting was opportunistic. While all pretest participants were fluent in the survey language being pretested, no other demographic characteristics were recorded. 

In the Discussion section we have also now expressly noted our approach to pretesting as a limitation (see line 400 of revised transcript). 

3. ETHICS 

Feedback: 

• Please provide additional details regarding participant consent. In the ethics statement in the Methods and online submission information, please ensure that you have specified whether consent was written or verbal/oral. If consent was verbal/oral, please specify: 1) whether the ethics committee approved the verbal/oral consent procedure, 2) why written consent could not be obtained, and 3) how verbal/oral consent was recorded. If your study included minors, please state whether you obtained consent from parents or guardians in these cases. If the need for consent was waived by the ethics committee, please include this information.

• Please include your full ethics statement in the ‘Methods’ section of your manuscript file. In your statement, please include the full name of the IRB or ethics committee who approved or waived your study, as well as whether or not you obtained informed written or verbal consent. If consent was waived for your study, please include this information in your statement as well. 

Response: 

The Methods section has been amended as shown below:

Ethics approval for pretesting was obtained (CUHREC Curtin University Human Research Ethics Committee 0395/2019-0395) and participants provided written consent. 

4. RESULTS

Feedback

• In line 141-142, "While identical independent translations were rare, many of the differences in translation were not material." What's the meaning of "material"?

Response:

We have amended the line as follows: “While identical independent translations were rare, many of the differences in translation were not material in the sense that they did not change the intended meaning of the source text.” 

5. DISCUSSION 

Feedback: 

• In the Discussion, it is stated that "Brislin's approach to back-translation was regarded as the gold standard". What's the justification for adopting other approach e.g. TRAPD for translation but not adopting Brislin's approach? What's the advantages of TRAPD over Brislin's approach?

Response: 

The Introduction has been revised as follows to address this earlier in the manuscript: 

It is generally recognised that ‘team translation’ represents the best current method for translating surveys.[11] The Guidelines for Best Practice in Cross-Culture Surveys recommend “a team translation approach for survey instrument production” noting that “[o]ther approaches, such as back translation, although recommended in the past, do not comply with the latest translation research” [11]. ‘Team translation’ is considered preferable to other approaches on the basis that it enables people with complementary knowledge and expertise (e.g. language and cultural experts, survey researchers, and people with expert knowledge relevant to the particular survey topic) to work together to arrive at the best translation to ensure that survey items convey what they were intended to convey.[12] In team translation, two independent translations are produced and are then compared (item-by-item) by bilingual reviewers who work with an adjudicator to identify the ‘best’ translation for pretesting.[11] However, there are relatively few published examples describing how team translation has been implemented in practice, and much of the available literature relates to translations carried out as part of large and relatively well-resourced surveys, such as the European Social Survey.[12-1613-17] 

The Discussion has also been revised as follows: 

Finally, it is worth remembering that our understanding of what constitutes best practice changes over time. It was not long ago that For decades, Brislin’s approach to back-translation was regarded as the gold standard.[32 12] However, an over-reliance on back-translation is now considered prone to producing more literal translations which can overlook important conceptual or cultural nuances.[12, 34, 35] As Bretschneider and colleagues note, “[t]he term ‘best practice’ implies that it is best when compared to any alternative course of action”.[33] While the in-depth discussions that occur in team translation methods such as TRAPD are considered useful in detecting nuances in meaning and achieving equivalence between languages, However it has been noted empirical data on “the contribution of the TRAPD process … to the overall quality of the translated survey instrument”[24] remains limited, and studies comparing the results of different translation methods are even more rare. Consequently, it makes sense for researchers to remain open to exploring and gathering data on a variety of approaches to translation provided they are transparently reported and critically appraised. 

6. STYLE

Feedback: 

Response: 

All files have been re-uploaded to meet the requirements for file naming. All other style requirements have been reviewed and the following changes have been made: 

• Line numbers added to title page; 

• Title on title page centred and font size increased; 

• Corresponding authors initials added in parentheses after email address; 

• Abstract reformatted to appear as a single paragraph with first-line indentation; 

• Figure 1 now cited as ‘Fig 1’; 

• Full stops added to the end of Table titles. 

7. FINANCIAL DISCLOSURE

Feedback: 

• We note that one or more of the authors is affiliated with the funding organization, indicating the funder may have had some role in the design, data collection, analysis or preparation of your manuscript for publication; in other words, the funder played an indirect role through the participation of the co-authors. If the funding organization did not play a role in the study design, data collection and analysis, decision to publish, or preparation of the manuscript and only provided financial support in the form of authors' salaries and/or research materials, please do the following:

a) Review your statements relating to the author contributions, and ensure you have specifically and accurately indicated the role(s) that these authors had in your study. These amendments should be made in the online form.

b) Confirm in your cover letter that you agree with the following statement, and we will change the online submission form on your behalf: 

“The funder provided support in the form of salaries for authors [insert relevant initials], but did not have any additional role in the study design, data collection and analysis, decision to publish, or preparation of the manuscript. The specific roles of these authors are articulated in the ‘author contributions’ section.

Response:

We confirm the following statement: 

The funder Curtin University provided support in the form of salaries for authors DV, MR, AR, and RL and scholarship support to NP, but did not have any additional role in the study design, data collection and analysis, decision to publish, or preparation of the manuscript. The specific roles of these authors are articulated in the ‘author contributions’ section.

8. FIGURES

Feedback: 

• Please remove your figures from within your manuscript file, leaving only the individual TIFF/EPS image files, uploaded separately. These will be automatically included in the reviewers’ PDF.

Response: 

• The figure has been deleted in the revised manuscript. 

We are grateful for the contributions you have made to help improve this paper. We hope that our revisions have adequately addressed the issues you have identified. 

Yours sincerely, 

The Authors

---

## [Decision Letter · Decision Letter 1]

4 Aug 2021

PONE-D-21-17384R1

Translating best practice into real practice: methods, results and lessons from a project to translate an English sexual health survey into four Asian languages

PLOS ONE

Dear Dr. Vujcich,

Thank you for submitting your manuscript to PLOS ONE. After careful consideration, we feel that it has merit but does not fully meet PLOS ONE’s publication criteria as it currently stands. Therefore, we invite you to submit a revised version of the manuscript that addresses the points raised during the review process.

ACADEMIC EDITOR:

Thank you for resubmitting the manuscript. The authors have attempted to address the comments, but the justification for some issues are not well supported, including the disproportion sample size for pre-testing, and the strengths of adopting TRAPD model for instrument translation. Suggest to discuss further on these two issues. Also discuss the impact and significance of this study on future research.

We look forward to receiving your revised manuscript.

Kind regards,

Ka Ming Chow

Academic Editor

PLOS ONE

Reviewers' comments:

Reviewer's Responses to Questions

**Comments to the Author**

1. If the authors have adequately addressed your comments raised in a previous round of review and you feel that this manuscript is now acceptable for publication, you may indicate that here to bypass the “Comments to the Author” section, enter your conflict of interest statement in the “Confidential to Editor” section, and submit your "Accept" recommendation.

Reviewer #1: (No Response)

2. Is the manuscript technically sound, and do the data support the conclusions?

Reviewer #1: Partly

3. Has the statistical analysis been performed appropriately and rigorously? 

Reviewer #1: N/A

4. Have the authors made all data underlying the findings in their manuscript fully available?

Reviewer #1: No

5. Is the manuscript presented in an intelligible fashion and written in standard English?

Reviewer #1: Yes

6. Review Comments to the Author

Reviewer #1: Thank you for revising and resubmitting the paper. The authors addressed the reviewer’s comments and provided supplementary information in the text to clarify several methodological issues. I understand that the study was limited by various methodological and budgetary constraints. However, some of the decisions in the study need to have a stronger justification. Please find my specific comments on the responses below.

1. Sample size: The authors have further explained the reason for recruiting a relatively small sample for pretesting, which is understandable. While budget is limited, I notice that the number of respondents is disproportionate among different communities. Only 3 Chinese (3.8% of the estimated population size) but 20 Vietnamese (66.7%) were recruited. This difference cannot be solely explained by budgetary constraint. Were there any practical reasons?

2. Discussion: The authors suggested additional research on the application of TRAPD model in translating survey instruments. Why kind of research or data do they recommend? Meanwhile, after revisiting the discussion, I have questions about the advantages of TRAPD model – (i) In this study, the ‘leaner’ TRAPD model could save time and resources for translation. However, is there evidence that the outcome of TRAPD model (e.g., culture adaptability and language) is comparable to the original TRAPD model and the conventional approach? (ii) Regarding the rigour, I notice that the TRAPD model was not adhered thoroughly. For example, some reviewers did not provide their comments for adjudication and documentation was simplified. Would the failure to adhere suggest the difficulty maintaining the rigour of the model in practice? (As you argued that the TRAPD model is the ‘real practice’)

Additional comments:

1. P.9: Please explain why simplified Chinese does not need to be reviewed separately? In fact, you mentioned that differences existed in Chinese used in Mainland China, Taiwan, and Hong Kong (in fact, there are other major communities using Chinese).

2. P.9-13: Were findings of all languages presented in the Results section? I notice that much of the results focuses on the Traditional Chinese version.

3. P.14-18: I think the numbering of paragraphs is not necessary, especially when some points are actually related (e.g., #6 and #7).

Overall, I would suggest the authors to review the aim of this article – Is it a methodological discussion or research paper? If it focuses on the methodology, a more critical discussion is warranted to compare the TRAPD model with other translation approaches in terms of its reliability and practicality, with sharing of practical experience and insightful recommendations. If it is more research-focused, findings need to be presented in an analytical manner in respective to a well-defined research aim.

7. PLOS authors have the option to publish the peer review history of their article (what does this mean?). If published, this will include your full peer review and any attached files.

Reviewer #1: No

---

## [Author Response · Author response to Decision Letter 1]

14 Nov 2021

November 2021

Dear Editors, 

Translating best practice into real practice: methods, results and lessons from a project to translate an English sexual health survey into four Asian languages 

Thank you for providing additional feedback on our manuscript. We have reproduced and responded to each item of feedback below (see attached version for proper formatting). Red font indicates new material, and ‘strike through’ shows where material from the previous draft has been deleted. 

Introduction

1. The authors have attempted to address the comments, but the justification for some issues are not well supported, including … the strengths of adopting TRAPD model for instrument translation. [Editor]

We amended the Introduction to further emphasise the strengths of team translation, one version of which is TRAPD: 

There are several approaches to translation. Forward-only translation (also known as ‘direct’ or ‘one-for-one’ translation) involves a single individual translating an instrument from one language (the source language) into a second language (the target language).[11] While forward-only translation has the advantage of saving time and costs, it is considered problematic because it “involves a total dependence on the [single] translator’s skill and knowledge, and often results in low validity and reliability”.[12] Forward-backward translation (also known simply as ‘back translation’) represents an attempt to overcome the risks inherent in relying on a single individual. In forward-backward translation, a second individual translates the target language instrument back into the source language; the original source language instrument and the back-translated source language instrument are then compared, and any discrepancies serve as indications of the need for further refinements of the translation.[13] However, a criticism of forward-backward translation is that it has the potential to focus too narrowly on the task of literal translation at the expense of ensuring that the translation captures the intended meaning of the survey item in a way that is clear and suitable for the intended audience.[14] Similarly, Behr cites an example in which ‘care services’ was forward translated into German as pflegedienste and back translated as ‘care services’ suggesting no error, when in fact the translated term “did not fit the questionnaire context since it is only used in the context of the ill and/or the elderly and is thus not fitting to general child care services”.[15] Ozolins and colleagues have reported how some forward translators choose literal translations (despite their misgivings as to whether it actually captures the intended meaning) because they do not want their translation to be flagged as an ‘error’ by the back translator.[16] 

Consequently, there has been a growing call for more nuanced and layered approaches to instrument translation in which accredited translators, other people who speak both the source and target languages, survey researchers, and subject matter experts work together to produce translated surveys which: (1) capture the intended meaning of the source instrument; (2) reflect the cultural and contextual specificities of the target population; and (3) will facilitate meaningful comparisons of data across populations.[13, 17-20] Indeed, the Guidelines for Best Practice in Cross-Cultural Surveys recommend “a team translation approach for survey instrument production” noting that “[o]ther approaches, such as back translation, although recommended in the past, do not comply with the latest translation research” [21]. ‘Team translation’ (also known as ‘committee translation’) is considered preferable to other approaches on the basis that it enables people with complementary knowledge and expertise (e.g. language and culture, survey methodology, knowledge relevant to the particular survey topic) to work together to arrive at the best translation to ensure that survey items convey what they were intended to convey to the target audience.[22] 

While team translation can assume a variety of forms, the approach known as TRAPD (Translation, Review, Adjudication, Pretesting, and Documentation) is the version endorsed in the Guidelines for Best Practice in Cross-Cultural Surveys.[21] Under the TRAPD model, In team translation two independent translations are produced and are then compared (item-by-item) by bilingual reviewers who possess study design and subject-matter knowledge, who and work with an adjudicator to identify the ‘best’ translation for pretesting; each step is documented for transparency.[21, 23] 

2. I would suggest the authors to review the aim of this article – Is it a methodological discussion or research paper? If it focuses on the methodology, a more critical discussion is warranted to compare the TRAPD model with other translation approaches in terms of its reliability and practicality, with sharing of practical experience and insightful recommendations. If it is more research-focused, findings need to be presented in an analytical manner in respective to a well-defined research aim. [Reviewer]

Thank you for this suggestion. We have revised the Introduction to more clearly communicate the intended aim and scope of our manuscript: 

In this article, we apply the TRAPD model to translate an English-language sexual health and blood-borne virus survey into four languages for migrants living in Australia. The aim of the study is to:

(1) document how TRAPD can be applied in practice, including any challenges in its application;

(2) provide examples of issues identified through the processes of team-based ‘review and adjudication’ and pretesting, which are key features of the TRAPD model; 

(3) offer guidance to future researchers who seek to use the TRAPD method; and

(4) provide recommendations for further research priorities on the subject of survey translation. 

The aim is not to empirically test the effectiveness of TRAPD relative to other methods of translation. 

The study makes an important contribution to the literature since there are However, there are relatively few published examples describing how TRAPD team translation has been implemented in practice the context of survey research, despite it being the model endorsed in the Guidelines for Best Practice in Cross-Cultural Surveys., and m Much of the available TRAPD literature relates to translations carried out as part of large and relatively well-resourced surveys, such as the European Social Survey.[24-28] However, as Sha and Lai have argued that “[i]t is important to identify a viable translation process that can be adapted and tailored to the varying level of expertise and resources available”.[29] The purpose of this paper is to document the process, results and lessons from a project to translate an English-language survey into four languages for migrants living in Australia. The project formed one part of study to ascertain the feasibility of conducting a periodic national survey in Australia to measure migrants’ knowledge, attitudes and practices in relation to sexual health and blood-borne viruses.

Amendments have also been made to the Results and Discussion sections of the paper to more explicitly reflect the aims described above. 

Methods

3. “The authors have further explained the reason for recruiting a relatively small sample for pretesting, which is understandable. While budget is limited, I notice that the number of respondents is disproportionate among different communities. Only 3 Chinese (3.8% of the estimated population size) but 20 Vietnamese (66.7%) were recruited. This difference cannot be solely explained by budgetary constraint. Were there any practical reasons?” [Reviewer; editor also requested justification for the disproportionate sample size]

We have provided the following additional explanation: 

The larger sample size for Vietnamese pretesting was opportunistic in the sense that a group of 20 participants were gathered for another purpose and expressed willingness to provide feedback on the translated instrument; although this resulted in the Vietnamese pretest sample being larger than those representing other language groups, the opportunity to obtain more feedback with minimal additional resources was recognised as an efficient means of obtaining more data to check instrument validity. There was no intention to engage in any statistical comparison of differences in pretest responses between the communities. 

4. Please explain why simplified Chinese does not need to be reviewed separately? In fact, you mentioned that differences existed in Chinese used in Mainland China, Taiwan, and Hong Kong (in fact, there are other major communities using Chinese).

As stated in the Introduction, the four languages for translation using the TRAPD method were Traditional Chinese, Karen, Khmer and Vietnamese. Translation into Simplified Chinese using the TRAPD method was not a stated aim of this research and, consequently, the results from the Simplified Chinese adaptation process are not presented in the Results section (or in Appendix S6). We have decided to remove the reference to Simplified Chinese in the Methods section to avoid confusion. The process of adapting the Traditional Chinese survey into Simplified Chinese was distinct from the TRAPD method and, as such, does not fall within the scope of the current paper; it is better suited to a separate manuscript on the subject of the process of adapting translated survey tools for use with different linguistic communities/dialects. 

Findings

5. In response to Question 4 (“Have the authors made all data underlying the findings in their manuscript fully available?”) the Reviewer has answered “No”. [Reviewer]

We note that we have provided raw data in the document marked Appendix 3 – TRAPD results. The results for each language are included in a separate workbook within the Excel file (screen shot in attached).

6. Were findings of all languages presented in the Results section? I notice that much of the results focuses on the Traditional Chinese version. [Reviewer] 

We note that the results for each language are included in a separate workbook within the Excel file marked Appendix 3 – TRAPD results (see above). We have also significantly amended the structure of the Results section, including the introduction of a new Table 5 (see in attached version) to highlight examples from all four languages. 

Discussion 

7. “I think the numbering of paragraphs is not necessary, especially when some points are actually related (e.g., #6 and #7).” [Reviewer] 

Thank you for this suggestion. We have now removed the numbering from the paragraphs. The Discussion section has been significantly redrafted in light of the feedback (point 2 above) that “a more critical discussion is warranted to compare the TRAPD model with other translation approaches in terms of its reliability and practicality, with sharing of practical experience and insightful recommendations.” 

8. “[A]fter revisiting the discussion, I have questions about the advantages of TRAPD model – (i) In this study, the ‘leaner’ TRAPD model could save time and resources for translation. However, is there evidence that the outcome of TRAPD model (e.g., culture adaptability and language) is comparable to the original TRAPD model and the conventional approach?” [Reviewer]

We have added the following paragraphs in our significantly reworked Discussion: 

Whereas much of the existing TRAPD literature reports on the use of the method in the context of large and relatively well-resourced, international surveys, this study documents the experience of applying the TRAPD method to a modestly-resourced survey in one country. The purpose of this study is not to provide evidence that TRAPD is the final word in effective translation methods. As has been noted elsewhere, empirical data on “the contribution of the TRAPD process … to the overall quality of the translated survey instrument”[18] remains limited, and more studies comparing the results of different translation methods are needed. Instead, this paper shows that the TRAPD method has value in identifying errors and exposing nuances in translation, but that it can be difficult to implement in practice, particularly where resources are low and translations into many languages are required. As Knight and colleagues acknowledge, the reality of studying minority populations is that “research context often makes full compliance with … best practice recommendations impossible and … the best one can do is to make incremental approximations of these recommendations.”[34] 

As was reported in the Methods, deviations from TRAPD ‘best practice’ occurred. Significantly, it was not possible to employ or compensate translation reviewers; instead, the task of reviewing was undertaken in-kind which made it difficult both to recruit multiple reviewers for each language, and to rigidly proscribe the manner in which the translation reviews were undertaken. For instance, while reviewers were instructed to code translation errors using the European Social Survey (ESS) Verifier Intervention Categories, few reviewers used the ESS categories, and some did not document any reasons for preferring one translation over another which made the process of adjudication difficult. Problems around documentation in the TRAPD review process have been noted in other studies, including the European Social Survey where it was noted that “maintaining documentation can be burdensome … The documentation provided by the country teams on the development of ESS translations was at times meagre.”[21] 

In addition to being time-consuming, the practice of coding translation errors may not be familiar to bilingual reviewers who do not necessarily possess a research background. Many of the distinctions between the categories – for instance, register/wording issues, grammar/syntax issues and minor linguistic defects (which can include grammar) – are subtle or require a high level of linguistic awareness. This suggests that the use of alternative coding protocols should be considered and tested, and efforts should be made to simplify the process of documenting reasons for decision at each stage of the translation process. In an effort to understand the difficulties they experienced in obtaining adequate documentation from translation partners, Harkness, Villar and Edwards surmised “[t]his may be because those involved were not familiar with how and what to document, but it is also likely that the effort involved in manual documentation played a role.”[21] One solution may be for a member of the research team to assume responsibility for documentation by sitting with each panel member (either physically or virtually) while they independently review the translations. The researcher could record the preferred translation for each item and the reviewer’s reasons, using probing questions where necessary; this was the approach taken for the Karen translation in our study. An alternative would be to investigate the development and use of specialised software to manage the documentation process. It is noted that an online, interactive Translation Management Tool was developed for the Survey of Health, Ageing and Retirement in Europe (SHARE) and tested in round 8 of the European Social Survey in 2016, although a number of areas for improvement were identified.[21] Currently, the tool is not publicly available but can be tailored to support specific survey requirements at a cost. 

It is not possible to determine whether closer adherence to TRAPD ‘best practice’ in this study would have improved the quality of the resulting translations. A literature review of studies using multi-step, team-based translations of health quality of life questionnaires found that “[a]lthough there is some evidence that different methods (i.e., ‘light/simpler’ vs. ‘heavy/complex’) yield similar results, this has been tested empirically only on a very limited scale” [12]. However, our study does demonstrate that even ‘light/simpler’ adaptations of the TRAPD method can successfully identify issues that may not have been apparent had non-team-based or single-round translation approaches been adopted. For instance: 

• relying on only one translator to conduct a forwards-only translation without further review would have resulted in important errors being overlooked (e.g. the Khmer translation of ‘casual sexual partner’ as ‘unprotected sexual partner’, and the Karen translation of ‘committed relationship’ as ‘willing in mind to live together’); 

• relying on forward-backward translation only may not have revealed importance nuances (e.g. the translation of hepatitis B as 乙型肝炎 would likely have been translated back into English as ‘hepatitis B’ suggesting no problem with the translation, when in fact the review process highlighted that it would not be inclusive of/familiar to Taiwanese-Chinese speakers); 

• not including researchers as partners in the translation process may have resulted in translations which did not reflect the intended meaning of the survey items (e.g. a bilingual reviewer without a sexual health and blood-borne virus background favoured the narrow, heteronormative translation of Karen translation of ‘sexual activities’ which meant ‘sex among men and women’); 

• pretesting assisted in identifying typographical/formatting errors made when incorporating the reviewed translations into the survey tools (e.g. the omission for the term ‘no/not’ in one item in the Karen survey, and parts of the Karen script not being displayed correctly); 

• the benefits of assembling a team comprising people with complementary expertise was particularly evident when attempting to translate technical, medical terminology in a way that, on the one hand, is meaningful to the target audience and, on the other hand, guards against perpetuating terms which are stigmatising or give rise to inaccurate assumptions (e.g. the difference between ‘clinical’ and ‘lay’ translations of HIV into Traditional Chinese). 

It is, of course, possible that some errors in translation were not detected as a result of the approach we adopted. For this reason, we have taken the rare step of making all of our translation data publicly available (Appendix S3)… 

9. “Regarding the rigour, I notice that the TRAPD model was not adhered thoroughly. For example, some reviewers did not provide their comments for adjudication and documentation was simplified. Would the failure to adhere suggest the difficulty maintaining the rigour of the model in practice? (As you argued that the TRAPD model is the ‘real practice’)” [Reviewer]

The Reviewer is correct to identify our conclusion that it may not be possible to rigidly apply TRAPD in all contexts. As noted in our response to Comment 8 (above), we argue that ‘light/simpler’ adaptations may be necessary and justifiable, and that more research needs to be undertaken on this point. 

10. “The authors suggested additional research on the application of TRAPD model in translating survey instruments. Why kind of research or data do they recommend?” [Reviewer]

11. “discuss the impact and significance of this study on future research.” [Editor] 

We have responded to Comments 10 and 11 through the addition of the following paragraphs: 

Given the dearth of clear empirical evidence about the most accurate and feasible method of undertaking translations, we encourage future researchers to follow our example. Greater transparency around survey translation research processes and results will: (1) allow the accuracy of different approaches to be studied in greater detail; (2) provide practical insight into effective ways of adapting established processes in response to different needs, settings and levels of resourcing; and (3) enable the validity of the survey results to be assessed with reference to the quality of the underlying translations. 

Unless a culture of transparent trial-and-error in translation is fostered, negative consequences will follow. First, survey researchers may feel daunted by the complexity (and uncertainty) of translation and may therefore avoid attempts to engage with linguistically diverse populations; this would either result in: (a) underrepresented sections of the population continuing to be overlooked in studies geared towards collecting data to improve health outcomes, or (b) respondents having no option but to complete surveys in a language other than their preferred language, thus increasing the potential for survey items to be misunderstood and data quality to be compromised. Second, survey researchers who have attempted to undertake translations but whose methods have deviated from ‘best practice’ (e.g. due to circumstantial challenges like budget limitations) may hesitate to publish their translation processes and results for fear of being criticised; the consequence of this would be a continued dearth of information about the most effective and efficient means of undertaking survey translations. As a recent editorial in Nature Human Behaviour makes plain, “[s]cience is messy, and the results of research rarely conform fully to plan or expectation. ‘Clean’ narratives are an artefact of inappropriate pressures and the culture they have generated.”[36] It is only through more detailed accounts of the processes and results of survey translation that we can come closer to understanding which methods (and variations of methods) are best suited to produce the most accurate results in specific contexts. Given the increase in the scale of international (pre-pandemic) migration,[37] and the growing appetite for multinational, multicultural and multiregional surveys,[38] the importance of continuing to build the evidence-base for survey translation is both apparent and urgent. 

We trust that these amendments adequately address your comments and we are grateful for the role you have played in strengthening our manuscript. 

Yours sincerely, 

The authors

---

## [Decision Letter · Decision Letter 2]

19 Nov 2021

PONE-D-21-17384R2Translating best practice into real practice: methods, results and lessons from a project to translate an English sexual health survey into four Asian languagesPLOS ONE

Dear Dr. Vujcich,

Thank you for submitting your manuscript to PLOS ONE. After careful consideration, we feel that it has merit but does not fully meet PLOS ONE’s publication criteria as it currently stands. Therefore, we invite you to submit a revised version of the manuscript that addresses the points raised during the review process.

Thank you for submitting the revised manuscript. Basically, the authors have adequately addressed all the comments. However, there are still some minor comments and suggestions for improving the paper which are listed as follows:

1. In the abstract, in Line 44 and 45, the sentence "...TRAPD method can identify issues that may not have been apparent had non-team-based or single-round translation approaches have been adopted" is difficult to understand. Please rephrase and revise.

2. In Line 116, the sentence "the aim is not to empirically test the effectiveness of TRAPD relative to other methods of translation." can be deleted.

3. I would like to clarify that supplementary files including S1, S2 and S3 will not be included in the published manuscript.

4. In the section or pre-testing, in Line 205-219, "The minimum target sample size ...... who would likely to speak English not well or not at all is unknown" should be deleted as the calculated sample size was not achieved and not related to the pre-testing survey.

5. In Table 4, it is shown that 9 questions resulted in identical independent translations, but in Line 249, it is stated that 12 were identical. Please clarify and revise accordingly. Please ensure that your decision is justified on PLOS ONE’s publication criteria and not, for example, on novelty or perceived impact.

We look forward to receiving your revised manuscript.

Kind regards,

Ka Ming Chow

Academic Editor

PLOS ONE

Journal Requirements:

Reviewers' comments:

Reviewer's Responses to Questions

**Comments to the Author**

1. If the authors have adequately addressed your comments raised in a previous round of review and you feel that this manuscript is now acceptable for publication, you may indicate that here to bypass the “Comments to the Author” section, enter your conflict of interest statement in the “Confidential to Editor” section, and submit your "Accept" recommendation.

Reviewer #1: All comments have been addressed

2. Is the manuscript technically sound, and do the data support the conclusions?

Reviewer #1: Yes

3. Has the statistical analysis been performed appropriately and rigorously? 

Reviewer #1: N/A

4. Have the authors made all data underlying the findings in their manuscript fully available?

Reviewer #1: Yes

5. Is the manuscript presented in an intelligible fashion and written in standard English?

Reviewer #1: Yes

6. Review Comments to the Author

Reviewer #1: Thank you very much for thoughtfully addressing our comments. The revised manuscript provides a critical reflection on the adoption of the TRAPD method to tranlate the sexual health and blood-borne virus survey. The revised Introduction is well-written and demonstrates the significance of using TRAPD method. In the Discussion, the authors successfully examined the processes, shared the lessons learnt, and compared the current practice with the original TRAPD method.

One suggestion is to add a brief conclusion at the end of the manuscript.

7. PLOS authors have the option to publish the peer review history of their article (what does this mean?). If published, this will include your full peer review and any attached files.

Reviewer #1: **Yes: **Marques Ng

---

## [Author Response · Author response to Decision Letter 2]

22 Nov 2021

Dear Editors, 

Translating best practice into real practice: methods, results and lessons from a project to translate an English sexual health survey into four Asian languages 

Thank you for providing additional feedback on our manuscript. We have reproduced and responded to each item of feedback below.

FEEDBACK FROM EDITOR

1. In the abstract, in Line 44 and 45, the sentence "...TRAPD method can identify issues that may not have been apparent had non-team-based or single-round translation approaches have been adopted" is difficult to understand. Please rephrase and revise.

Both the unmarked and tracked versions of the manuscript had the correct wording, as follows: 

However, our study does demonstrate that even adaptations of the TRAPD method can identify issues that may not have been apparent had non-team-based or single-round translation approaches been adopted.

2. In Line 116, the sentence "the aim is not to empirically test the effectiveness of TRAPD relative to other methods of translation." can be deleted.

We have deleted the sentence. 

3. I would like to clarify that supplementary files including S1, S2 and S3 will not be included in the published manuscript.

We understand from PLOS ONE guidelines that:

Supporting information is auxiliary to the main content of the article. In the online version of the published article, readers access the files via hyperlinks in the Supporting Information section of the article. PLOS hosts these files on its servers and also deposits them on Figshare to increase compliance with the FAIR principles of data sharing. Supporting information files are published exactly as provided, and are not copyedited … The PLOS publishing platform supports any file type for supporting information files. 

We are happy for the supplementary files to be accessible via hyperlinks to the documents hosted on the PLOS servers/Figshare. However, if you require us to provide the files in a different format, or to host them on our project website, please let us know so that we can accommodate your request. 

4. In the section or pre-testing, in Line 205-219, "The minimum target sample size ...... who would likely to speak English not well or not at all is unknown" should be deleted as the calculated sample size was not achieved and not related to the pre-testing survey.

We have deleted these lines as indicated in the attached response to reviewers, and amended reference numbers accordingly.

5. In Table 4, it is shown that 9 questions resulted in identical independent translations, but in Line 249, it is stated that 12 were identical. Please clarify and revise accordingly.

Thank you for pointing this out; it should read ‘nine’; we have amended accordingly. 

FEEDBACK FROM REVIEWER 1 

6. One suggestion is to add a brief conclusion at the end of the manuscript.

We have added the following sub-section: 

Conclusions

Survey translation is time-consuming and costly and, as a result, deviations from TRAPD ‘best practice’ occurred. It is not possible to determine whether closer adherence to TRAPD ‘best practice’ would have improved the quality of the resulting translations. However, our study does demonstrate that even adaptations of the TRAPD method can identify issues that may not have been apparent had non-team-based or single-round translation approaches been adopted. Given the dearth of clear empirical evidence about the most accurate and feasible method of undertaking translations, we encourage future researchers to follow our example of making translation data publicly available to enhance transparency and enable critical appraisal.

We trust that these amendments adequately address your comments. Please advise whether you require anything further from us. 

Yours sincerely, 

The authors

---

## [Editor Report · Decision Letter 3]

24 Nov 2021

Translating best practice into real practice: methods, results and lessons from a project to translate an English sexual health survey into four Asian languages

PONE-D-21-17384R3

Dear Dr. Vujcich,

We’re pleased to inform you that your manuscript has been judged scientifically suitable for publication and will be formally accepted for publication once it meets all outstanding technical requirements.

Kind regards,

Ka Ming Chow

Academic Editor

PLOS ONE

---

## [Editor Report · Acceptance letter]

29 Nov 2021

PONE-D-21-17384R3 

Translating best practice into real practice: methods, results and lessons from a project to translate an English sexual health survey into four Asian languages 

Dear Dr. Vujcich:

I'm pleased to inform you that your manuscript has been deemed suitable for publication in PLOS ONE. Congratulations! Your manuscript is now with our production department. 

Kind regards, 

on behalf of

Dr. Ka Ming Chow 

Academic Editor

PLOS ONE